# Long-Term Outcomes following Common Femoral Endarterectomy

**DOI:** 10.3390/jcm11226873

**Published:** 2022-11-21

**Authors:** Takuya Hashimoto, Satoshi Yamamoto, Masaru Kimura, Masaya Sano, Osamu Sato, Juno Deguchi

**Affiliations:** 1Department of Vascular Surgery, Saitama Medical Center, Saitama Medical University, Kawagoe 350-0844, Japan; 2Department of Surgery, Ome Municipal General Hospital, Ome 198-0042, Japan

**Keywords:** peripheral artery disease, revascularization, common femoral endarterectomy

## Abstract

Thromboendarterectomy of the common femoral artery (CFA) for occlusive disease is a crucial procedure in vascular surgery. As an outcome reference for emerging endovascular procedures and new devices, we need more robust evidence of the outcome of this gold standard technique. The purpose of this study was to report 10-year results after femoral endarterectomy (FEA). A retrospective review of medical records at our institution identified eighty consecutive patients (91 limbs) who underwent FEA for CFA lesions. Indications for FEA included 50 limbs (55%) for intermittent claudication (IC) and 39 limbs (43%) with chronic limb-threatening ischemia (CLTI). Two limbs (2%) underwent FEA to prevent hemodynamic steal during extra-anatomical bypass. Adjunctive procedures included endovascular therapy in 32%. CFAs were closed with patch angioplasty in 44%. With a mean follow-up period of 39 months, the survival rates at 3 and 8 years were 85% and 77%, respectively. Limb salvage rates were 92% and 87%. Primary patencies were 98% and 84%. Freedom from target lesion revascularization was 95% at 3 years and 91% at 8 years. Our findings support the durability of FEA, with comparable long-term procedural results in CLTI patients as well as IC patients. Since the FEA is a gate maneuver for hybrid revascularization in CLTI patients, our findings support a strategy combining open and endovascular approaches. Femoral endarterectomy remains a durable solution for common femoral occlusive disease in IC and CLTI in the era of endovascular therapy.

## 1. Introduction

Surgical revascularization has been the mainstay for the treatment of ischemic limbs. To cure arterial obstruction, procedural steps for incision of the artery, extraction of the thrombus, and closure of the vessel are straightforward ways to revascularize ischemic limbs. This concept of thromboendarterectomy was promulgated as early as the 1880s [1]. After early attempts failed, presumably due to a lack of anticoagulants, the first successful case was reported in 1946 after the clinical use of heparin. Since then, femoral endarterectomy (FEA) has been a crucial procedure in vascular surgery, and current guidelines strongly recommend FEA, even in this endovascular era. As a reference for emerging endovascular procedures and new devices, we need robust evidence on the long-term outcome of this gold standard procedure. Several studies have demonstrated the utility of this procedure [2,3,4]. However, further evidence of this procedure’s long-term efficacy is still warranted, especially for chronic limb-threatening ischemia (CLTI), the most devastating mode of peripheral artery disease (PAD), because this procedure is now frequently used as a gate maneuver for hybrid revascularization in those patients. Accordingly, we sought to clarify 10-year outcomes after FEA. Future directions of hybrid revascularization for PAD and the role of FEA as a gate maneuver are discussed.

## 2. Materials and Methods

This retrospective, single-center study investigated long-term clinical results following FEA. The study included all consecutive patients who underwent FEA with the exclusion of FEAs performed at bypass anastomosis sites and emergency cases presenting acute limb ischemia. Preoperative patient background data included age, sex, chronic kidney disease necessitating regular hemodialysis, hypertension, diabetes mellitus, and coronary artery disease. Tobacco use, medications, and symptoms for operative indications were also recorded.

### 2.1. Operational Technique

FEA was performed using the standard technique. The CFA and the distal end of the external iliac artery were exposed and mobilized. The distal extent of the dissection, i.e., within the CFA or into the superficial femoral artery (SFA) or deep femoral artery (DFA), was determined by preoperative assessment and intraoperative findings. Under systemic heparinization, arterial clamp and longitudinal arteriotomy were made along the diseased lesion, and occluding plaques were dissected and removed at the appropriate dissection plane. The distal end of the intima was reattached with interrupted sutures. The arteriotomy was closed with 6-0 sutures in cases of sufficient CFA diameter. In patients with small diameter CFAs or arteriotomy into SFA or DFA, patch angioplasty was performed to prevent narrowing after closure. For hybrid procedures with EVT, FEA was typically performed before the endovascular component. To achieve complete revascularization in CLTI cases, we generally prefer tibial and pedal bypass surgery using autogenous veins over EVT as a distal procedure.

Operative details included anesthesia, use and type of patch angioplasty, and adjunctive procedures. Technical success of FEA and thirty-day morbidity and mortality are described. We analyzed overall survival rates, limb salvage, primary patency, and freedom from target lesion revascularization as primary endpoints. Postoperative surveillance protocol included clinical assessment and noninvasive hemodynamic tests. Imaging studies, mainly Doppler ultrasonography and computed tomographic angiography, of the treated arterial segment or adjunctive bypass graft were combined as appropriate. Surveillance continued at 3 and 6 months, and every 6 to 12 months after that, in a manner compatible with recent recommendations [5].

### 2.2. Statistical Analysis

All data were analyzed using Prism 8 software (GraphPad Software, Inc., La Jolla, CA, USA). We calculated percentages for categorical variables and means ± standard deviations or medians (ranges) for continuous variables. The Kaplan–Meier method was used to analyze rates of primary endpoints. Comparisons of group survival curves were obtained with the Log-rank (Mantel–Cox) test.

## 3. Results

### 3.1. Patient’s Background

Between April 2005 and March 2022, 91 FEAs were performed at Saitama Medical Center, Kawagoe, Japan, on 80 symptomatic patients with occlusive disease of the CFA. Table 1 summarizes patient demographics, comorbidities, medications, and symptoms of limbs.

Demographics and comorbidities were unremarkable for the current typical PAD patient population in Japan. Patients presented with diabetes mellitus (60.0%), coronary artery disease (43.4%), and chronic kidney disease on hemodialysis (51.3%). Former and current smokers comprised 63.8% of these patients. Fifty FEAs (54.9%) were performed on limbs with indications for intermittent claudication and 39 (42.9%) for CLTI. Two limbs (2.2%) underwent FEA to prevent hemodynamic steal in inflow-side limbs during extra-anatomical bypass. Medication included antiplatelet agents (96.3%), anticoagulants (10.0%), and statins (35.0%). The inclusion number of limbs per year is provided in Figure 1.

### 3.2. Operative Data

Table 2 summarizes the operative details of 91 procedures. Most FEAs were performed under general anesthesia (85.7%). CFAs were closed with patch angioplasty in 40 limbs (44.0%). Patch materials used were a Dacron strip in 32 limbs (80.0% among patch cases), bovine pericardium in 7 limbs (17.5%), and an SFA rotational flap in 1 limb (2.5%). Technical success was achieved in 100% of these procedures.

Adjunctive vascular procedures performed simultaneously with FEA are shown in Table 3. The total number of patients who required ipsilateral adjunctive vascular procedures was 50 (62.5%). Bypass of any type was performed in 24 limbs (26.3%). Endovascular therapy was performed in 29 limbs (31.9%). Among these, 20 limbs (69.0% of EVT cases) underwent inflow revascularization proximal to the CFA at the iliac arteries. Seven limbs (24.1%) underwent EVT at a site distal to the CFA, i.e., the concurrent femoral and popliteal lesions. Hybrid procedures with bidirectional EVT to the iliac and SFA-Pop lesions were done in two limbs (6.9%). 

### 3.3. Thirty-Day Morbidity and Mortality

Postoperative 30-day complications are reported in Table 4. The mortality rate was 2.2%. The cause of two perioperative deaths was rectal bleeding and arrhythmia, leading to multiple organ failure. Both occurred in CLTI patients on hemodialysis. Cardiac, pulmonary, and gastrointestinal complications were 4.4%, 4.4%, and 2.2%, respectively. Wound complications at the surgical incision of the groin were seen in two cases (2.2%).

### 3.4. Ten-Year Outcomes after FEA

Kaplan–Meier analysis revealed that survival rates 3 and 8 years after FEA were 85% and 77%, respectively. Limb salvage rates were 92% and 87%. Primary patencies were 98% and 84%. Freedom from target lesion revascularization was 95% at 3 years and 91% at 8 years (Figure 2). The mean observation period was 39.2 months.

Figure 3 shows the ten-year outcome stratified by initial limb status. A comparison of survival curves between IC and CLTI revealed superior survival (*p* = 0.0010) and limb salvage rate (*p* = 0.0002) in IC patients over CLTI patients. Primary patency (*p* = 0.8992) and freedom from TLR (*p* = 0.3182) did not show significant differences between groups. Furthermore, in order to clarify the contribution of comorbidities and medication status, survival curves were compared with the log-rank test. Hemodialysis (*p* = 0.5541), diabetes mellitus (*p* = 0.0934), smoking (*p* = 0.0900), and statin therapy (*p* = 0.3635) did not impact TLR.

## 4. Discussion

In this study, we demonstrated that FEA for common femoral occlusive disease has excellent clinical efficacy in the long term. Eight-year limb salvage and freedom from target lesion revascularization rates were 87% and 91% in our cohort of 43% CLTI. Previous studies on FEA have also shown favorable results similar to those seen here [2,3,4,6]. Kang et al. reported a 5-year freedom from reintervention rate of 78% (32% CLI) [2]. More recently, Kuma et al. reported a 5-year limb salvage rate of 95% (36% CLI) [4]. These studies demonstrate that FEA is a durable procedure to treat CFA occlusive disease.

Inherent operative risks of FEA include surgical site bleeding and wound complications compared with EVT. In our series, we experienced no postoperative bleeding and minimal complications at the wound site of FEA, demonstrating the safety of FEA, in addition to its excellent long-term results. However, increased use of dual antiplatelet agents for coronary artery disease could potentially hamper the safety of current results. On the other hand, developing negative pressure therapy at closed wounds may decrease wound complication rates, influencing future patient selection for FEA or EVT of CFA.

One of the prominent features of our study cohort is the high prevalence of hemodialysis (51.3%) compared with previous CFA studies, although this is now typical of CLTI patients in Japan. On the other hand, hemodialysis was not a significant risk factor for limb loss and target lesion revascularization of CFA in our series. Further analysis of such patient risk factors on outcomes in the CLTI population is warranted to improve patient selection and shared decision-making [7].

The role of FEA differs in IC and CLTI patients, and the results of FEA are affected by initial limb status. First, in cases of FEA for claudication, run-off status of outflow vessels impacts long-term outcomes. However, anatomical factors affecting FEA results for CLTI have not been examined thoroughly so far. Our study showed comparable freedom from target lesion revascularization in CLTI and IC patients in the presence of adjunctive distal procedures to achieve adequate run-off in CLTI cases. Second, the dissection plane during plaque removal is not known in FEA in CLTI, especially for patients with diabetes mellitus and on hemodialysis who are prone to Monckeberg’s medial sclerosis [8]. The severity of calcification and its distribution in the arterial layer supposedly differs between patients with claudication and those with CLTI. 

Multilevel complex lesions involving CFA in PAD patients are increasingly treated using hybrid procedures combining surgical endarterectomy and EVT [9,10]. During these procedures, FEA with patch angioplasty provides convenient access for simultaneous EVT of inflow and outflow lesions. The bovine pericardial (BP) patch is a newly added tool to treat complex femoral occlusive lesions. Whereas BP patches had been widely used after arteriotomies during cardiovascular surgery for three decades, they became available for femoral use in Japan in 2020. BP patches have several advantages compared to conventional patches. They are freely cut and shaped, are easy to handle, cause less suture bleeding, and are time-saving as well as vein-saving compared with vein patches [11] (Figure 4). Notably, intraoperative punctures are more accessible than Dacron patches and possibly safer than vein patches [12]. This property will enhance the use of BP patches during hybrid procedures for PAD. In our institution, we changed the patch material from Dacron to BP in 2020. Our initial results in seven cases with BP patches were favorable, without related complications. We believe that BP patches are promising to simplify FEA, profundaplasty, and concurrent iliac EVT, all proven long-term therapies for PAD.

In this era of emergent endovascular therapy, attention is shifting to reintervention and long-term clinical outcomes. Although EVT has become the first-line therapy for atherosclerotic disease of aortoiliac and femoropopliteal lesions, there are limited data to support EVT for CFA atherosclerosis [13]. Registry-based data used currently for the creation of objective performance goals (OPGs) usually do not provide full mid-term outcomes [14,15]. As a result, it is challenging to set OPGs even at 3-years, which may hamper evidence-based guidelines that genuinely benefit the sustainability of PAD patient limb and life. In contrast, well-conducted retrospective studies based on prospectively maintained databases provide essential information regarding the long-term durability of index procedures. Meta-analyses of these collected data could help refine the quality of OPGs for emerging endovascular therapies. Our local experience ascertained the extended durability of FEA as a gold-standard procedure, adding evidence for future OPGs and practice benchmarks.

The current study has several limitations, partly due to its single-center, retrospective design. Owing to the small sample size, we did not have robust data beyond mid-term, especially for CLTI patients whose survival was limited. Despite our relatively longer observation period compared with preceding studies [3,4,16,17], the follow-up rate was not perfect, reflecting real-world practice. Finally, the current study is not free from selection bias since our data were limited to those from patients with PAD who underwent FEA.

In conclusion, our findings add evidence to the durability of FEA as the standard open vascular procedure for IC and CLTI, maintaining a high bar for emerging endovascular therapies and devices. Further research should focus on developing longer-term OPGs for essential vascular procedures. Originally, FEA was the procedure for focal vascular reconstruction to treat claudication. Here, we demonstrated that the long-term procedural results in CLTI patients, i.e., primary patency and freedom from TLR, are comparable to those in IC patients, with certain conditions providing appropriate inflow and outflow via adjunctive procedures. Since FEA is a gate maneuver of hybrid revascularization in CLTI patients, our finding provides a rationale to support a strategy of combining open and endovascular approaches. Customized use of open and endovascular hybrid therapies according to patient conditions, limb status, and anatomical disease factors will allow PAD patients to achieve sustainable limb and life.

## Figures and Tables

**Figure 1 jcm-11-06873-f001:**
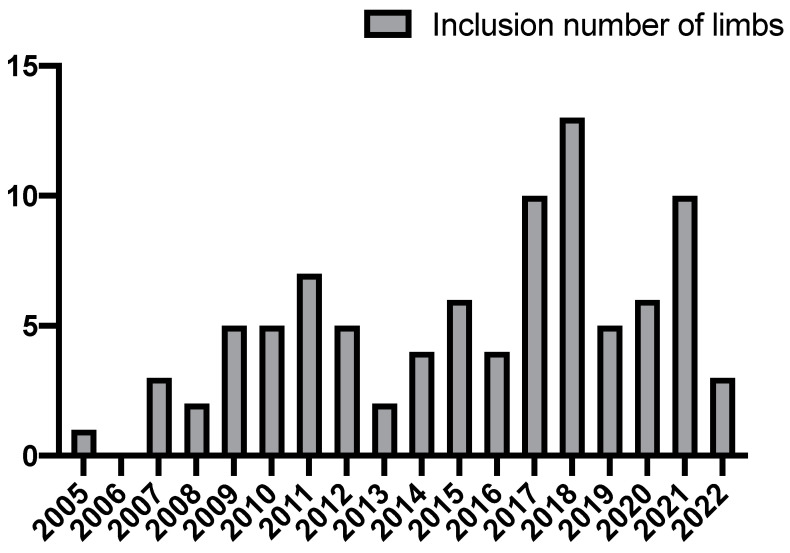
Inclusion number of limbs per year.

**Figure 2 jcm-11-06873-f002:**
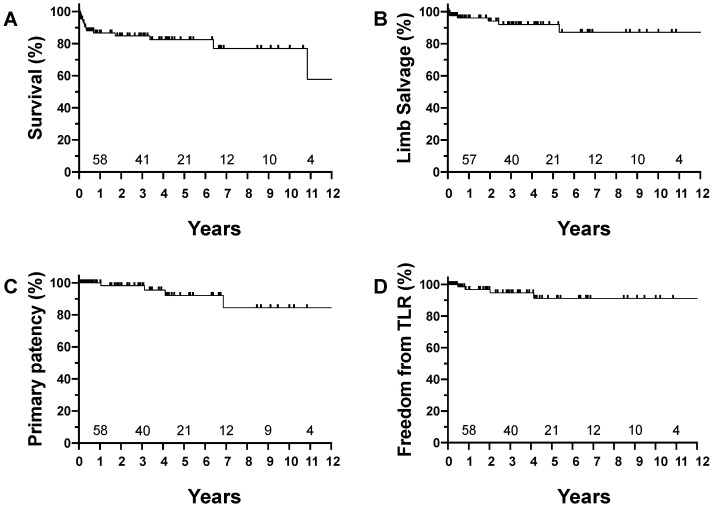
Ten-year outcome after femoral endarterectomy. (**A**) Survival; (**B**) limb salvage; (**C**) primary patency; (**D**) freedom from target lesion revascularization. Numbers at risk are shown above the x-axis.

**Figure 3 jcm-11-06873-f003:**
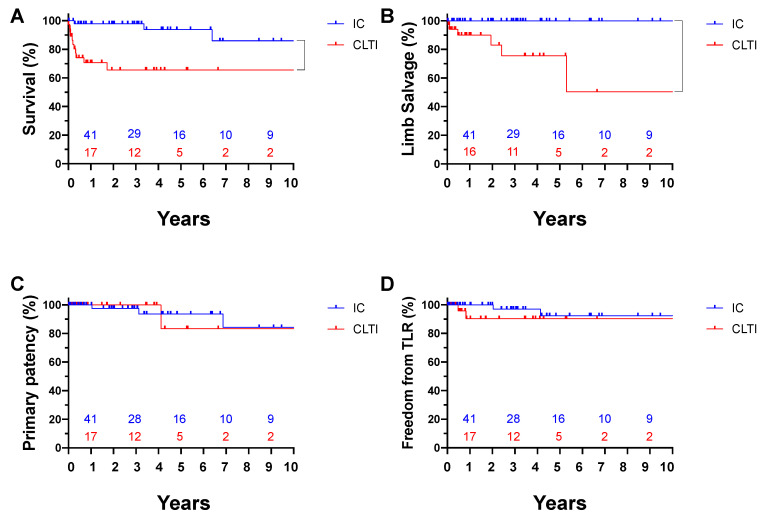
Ten-year outcomes after femoral endarterectomy, according to initial limb status. (**A**) Survival, *p* = 0.0010; (**B**) limb salvage, *p* = 0.0002; (**C**) primary patency, *p* = 0.8992; (**D**) freedom from target lesion revascularization, *p* = 0.3182. IC, intermittent claudication; CLTI, chronic limb-threatening ischemia. Numbers at risk are shown above the x-axis.

**Figure 4 jcm-11-06873-f004:**
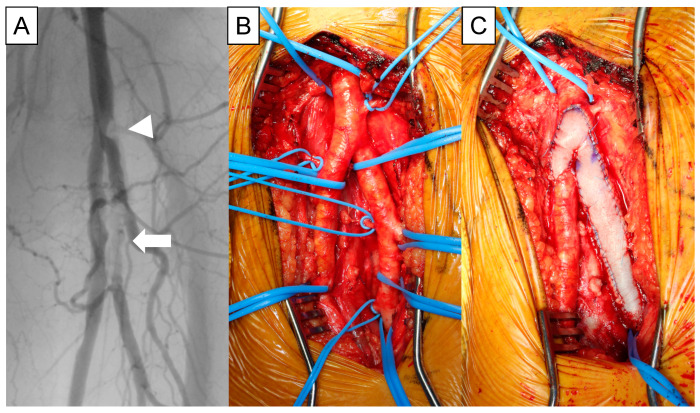
Common femoral endarterectomy with extended profundaplasty using a bovine pericardial patch. (**A**) Trans-arterial angiography of the left femoral artery. Stenosis at common femoral artery (arrowhead) and occlusion of deep femoral artery (arrow). (**B**) Intraoperative picture of the left femoral artery. (**C**) Reconstruction by patch-angioplasty using a bovine pericardial patch.

**Table 1 jcm-11-06873-t001:** Clinical background of the patients.

Clinical Parameters	Total 80 Patients, 91 Limbs (%)
Demographics (n = 80 patients)	
Age (years)	70.8 ± 10
Male	72 (90.0)
Comorbidity (n = 80 patients)	
Chronic kidney disease on hemodialysis	41 (51.3)
Diabetes mellitus	48 (60.0)
Coronary artery disease	35 (43.4)
History of tobacco use	51 (63.8)
Symptoms and indications (n = 91 limbs)	
Claudication	50 (54.9)
CLTI	39 (42.9)
Others ^1^	2 (2.2)
Medication (n = 80 patients)	
Antiplatelet agents	77 (96.3)
Anticoagulants	8 (10.0)
Statins	28 (35.0)

^1^ Prevention of hemodynamic steal after extra-anatomical bypass. CLTI, chronic limb-threatening ischemia.

**Table 2 jcm-11-06873-t002:** Operative details of femoral endarterectomy.

	Total 91 Limbs (%)
Anesthesia	
General	78 (85.7)
Epidural/spinal	8 (8.8)
Regional Nerve Block	1 (1.1)
Local	4 (4.4)
Closure of CFA	
Primary closure	51 (56.0)
Patch angioplasty	40 (44.0)
Dacron	32 (80.0 *)
Bovine pericardium	7 (17.5 *)
Rotational flap using SFA	1 (2.5 *)

* Percentage within patch angioplasty. CFA, common femoral artery; SFA, superficial femoral artery.

**Table 3 jcm-11-06873-t003:** Adjunctive vascular procedures of femoral endarterectomy.

	Total 91 Limbs (%)
Profundaplasty	13 (14.2)
Bypass	24 (26.3)
Cross over	4 (16.7 *)
SFA-popliteal artery	10 (41.7 *)
Distal bypass	10 (41.7 *)
EVT	29 (31.9)
Proximal	20 (69.0 ^§^)
Distal	7 (24.1 ^§^)
Bidirectional	2 (6.9 ^§^)
Contralateral FEA	10 (11.0)

* Percentage within bypass; ^§^ percentage within EVT.

**Table 4 jcm-11-06873-t004:** Perioperative complications of common femoral endarterectomy.

	Total 80 Patients, 91 Limbs (%)
Mortality	2 (2.2)
Gastrointestinal bleeding	1 (1.1)
Cardiac	1 (1.1)
Morbidity	10 (11.0)
Gastrointestinal	2 (2.2)
Pneumonia	4 (4.4)
Cardiac	4 (4.4)
Wound complication	2 (2.2)

## Data Availability

Not applicable.

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
