# Peer review of "Long-Term Outcomes following Common Femoral Endarterectomy"

_jcm, 2022, doi:10.3390/jcm11226873_

Round 1

Reviewer 1 Report

The authors presented the long-Term Outcomes Following Common Femoral Endarterectomy. The study included 80 patients accross 10 years. The topic is interest and the design of the study is good. some suggestions:

1. The complications like DM,hyperlipidemia, smoking..... contributes a lot to the results. It is neccessary to make more detailed analysis on the impact of those complicaitons.

2.Anti-platelets therapy is also associated with the results, as well as statins therapy. it seems this study has not pay much attention on them.

3.It is helpful to publish some pictures of the surgery

4.It is better to present their conclusions in the abstract

Reviewer 2 Report

Thank you for the opportunity to review the paper entitled: “Long-Term Outcomes Following Common Femoral Endarterectomy” by Hashimoto and colleagues, I enjoyed reading this paper, it provides important results regarding therapeutic options and the long-term basis for patients presenting with peripheral arterial disease. Perhaps not a novel topic but surely an important paper which may prove very useful for other centers. I do not have any other comments, A well put together study. 

Reviewer 3 Report

Hashimoto and colleagues, in their article "Log-Term Outcomes Following Common Femoral Endarterectomy," describe long-term survival up to 10 years in their single-center cohort after common FEA.

They used a single-center retropsective approach.

Despite the study design and appropriate presentation, the article has some weaknesses.

1. FEA is a well-studied, well-known technique that has been widely used for several decades. It is unclear what the new content of this article is. The authors should highlight this in their article.

2. How many patients had the potential to reach 10-year follow-up? Can the authors provide the inclusion numbers per year?

3. What was the cause of death and was it related to the procedure/vascular disease?

4. Please provide months or years instead of days (e.g., line 123 or figure one).

5. What was the total number of patients who required an adjunctive vascular procedure?

6. Please insert in figures "A-D" as used in the figure description.

7. For clarity, the "Operational Technique" and "Statistical Analysis" sections of the "Methods" section should be separated by headings.

Round 2

Reviewer 3 Report

Thank you very much for the improvment. I have no furhter comments on the content.

Please just take a second look to the presentation of the figure. Figure 2 (10-yr outcome after femoral endarterectomy) is tilted Figure 1. In my version the new and old Figure of the Kaplan-Meier analysis are overlayed. Please check this again. 

Author Response

Thank you for pointing this out. Now I have amended the figure number. New and old figures overlay should disappear when choosing to show only the final version and not to show track change on a Word file.